# New Financial Ratios Based on the Compositional Data Methodology

Salvador Linares-Mustarós [1], Maria Àngels Farreras-Noguer [1], Núria Arimany-Serrat [2] and Germà Coenders [3,*]

1 Department of Business Administration, University of Girona, 17003 Girona, Spain
2 Department of Economics and Business, University of Vic-Central University of Catalonia, 08500 Vic, Spain
3 Department of Economics, University of Girona, 17003 Girona, Spain
* Correspondence: germa.coenders@udg.edu; Tel.: +34-972418736

**Abstract:** Due to the type of mathematical construction, the use of standard financial ratios in studies analyzing the financial health of a group of firms leads to a series of statistical problems that can invalidate the results obtained. These problems originate from the asymmetry of financial ratios. The present article justifies the use of a new methodology using Compositional Data (CoDa) to analyze the financial statements of an industry, improving analyses using conventional ratios, since the new methodology enables statistical techniques to be applied without encountering any serious drawbacks, such as skewness and outliers, and without the results depending on the arbitrary choice as to which of the accounting figures is the numerator of the ratio and which is the denominator. An example with data on the wine industry is provided. The results show that when using CoDa, outliers and skewness are much reduced, and results are invariant to numerator and denominator permutation.

**Keywords:** financial ratios; management ratios; financial statement analysis; accounting ratios; compositional data (CoDa); financial performance; sectoral analysis; log ratio

**MSC:** 62P05; 91B99; 62F35; 62H99

## 1. Introduction

While standard financial ratios help to evaluate the financial statements of firms and corporations at an individual level accurately [1–4], unfortunately, when they are used as variables in statistical analyses of the financial health of an industry, the reliability of the results cannot be accepted as valid given that, as this article will show, the loss of symmetry produced when constructing standard financial ratios causes significant distortions when diagnosing financial health. Although this asymmetry in financial ratios has been known for some time [5–8], it has not received due attention from the area of accounting, despite the serious problem in which it casts doubt on the results of multiple studies that use standard ratios as variables [9–14]. The statistical problems of ratios have also been reported in other scientific fields [15]. The present article aims to correct this fact by providing a detailed explanation of the reason for this problem, which is also the root cause of other identified and related problems, such as the emergence of spurious outliers [16–20], and by showing how the results obtained are incoherent if ratios where the numerator and the denominator are permuted are used [16,21,22].

In the first part of this article, a mathematic counterexample with artificial data is used to evidence the serious drawbacks of using standard financial ratios in statistical studies at the industry level. Once the danger of incurring serious methodological problems when using standard financial ratios in this type of study is explained, a new type of financial ratio based on the methodology of compositional data analysis, or simply Compositional Data (CoDa), is suggested, the validity of the results of which has already been extensively tested in other fields [23–28]. While the CoDa methodology emerged from the fields of geometry and chemistry at the end of the last century [23,29], it has since been extended to

all the other scientific fields of study, including economics and other social sciences [30], and has started to be regularly used in studies in the area of finance [31–43] and, more recently, in the area of accounting [16,22,44–48]. The article uses a second example with real data of a particular industry in a European country, which shows the invalidity of the results obtained using the traditional methodology and how the proposed methodology avoids the abovementioned problems.

The contributions of this article are to show that the CoDa approach can be used to advantage in the accounting field, ensuring the validity of results, and to provide the research community in finance and accounting with a reasoned case for using a new methodology to analyze the financial statements in an industry.

Based on the line of argument presented, this article is organized in three main sections. Section 2 presents the artificial data example and focuses on showing the serious problems arising from the use of standard ratios in studies at the industry level, which result from the lack of symmetry of ratios and can lead to the invalidity of the results of the analysis. Financial ratios based on the CoDa methodology are presented in Section 3 as possible candidates to replace standard or conventional financial ratios. Section 4 presents the real data example comparing the use of conventional and compositional ratios, which aims to show the need to change from the usual working methodology in studies based on standard financial ratios by exposing the significant discrepancies in the two sets of results. Sections 5 and 6 present the discussion and conclusions, respectively.

## 2. Theory: The Problem of the Asymmetry of Standard Ratios and the Appearance of Spurious Outliers

This section explains in detail why, from a methodological point of view, the results of studies that use standard financial ratios as variables in statistical analyses of an industry cannot be considered valid. To this end, the starting point is an example with the simulated data of a group of ten firms, the values of two accounting magnitudes of which are given in the first two columns of Table 1.

**Table 1.** Artificial data of a ten-firm industry.

|  | Magnitude 1 | Magnitude 2 | $\alpha$ | $\dfrac{\text{Mg } 2}{\text{Mg } 1}$ | $\dfrac{\text{Mg } 1}{\text{Mg } 2}$ |
|---|---|---|---|---|---|
| Firm $E_1$ | 0.5 | 4 | ~82.875 (=45 + 37.875) | 8 | 0.125 |
| Firm $E_2$ | 1.5 | 3 | ~63.435 (=45 + 18.435) | 2 | 0.5 |
| Firm $E_3$ | 1.5 | 2.5 | ~59.035 (=45 + 14.035) | $1.\hat{6}$ | 0.6 |
| Firm $E_4$ | 1.8 | 3 | ~59.035 (=45 + 14.035) | $1.\hat{6}$ | 0.6 |
| Firm $E_5$ | 1.5 | 1.5 | ~45 (=45 + 0) | 1 | 1 |
| Firm $E_6$ | 3 | 3 | ~45 (=45 − 0) | 1 | 1 |
| Firm $E_7$ | 3 | 1.8 | ~30.965 (=45 − 14.035) | 0.6 | $1.\hat{6}$ |
| Firm $E_8$ | 2.5 | 1.5 | ~30.965 (=45 − 14.035) | 0.6 | $1.\hat{6}$ |
| Firm $E_9$ | 3 | 1.5 | ~26.565 (=45 − 18.435) | 0.5 | 2 |
| Firm $E_{10}$ | 4 | 0.5 | ~7.125 (=45 − 37.875) | 0.125 | 8 |

Given that this section focuses on the problems related to asymmetry, Magnitude 1 and Magnitude 2 in Table 1 are represented graphically in Cartesian coordinates. An examination of Figure 1 shows that the magnitudes recorded in Table 1 are chosen because they are symmetrical with respect to the 45° angle in the computation of the angles $\alpha$ of the rays that start at the origin and pass through the points, as shown in Figure 2. This symmetry is reflected in the column headed $\alpha$ in Table 1.

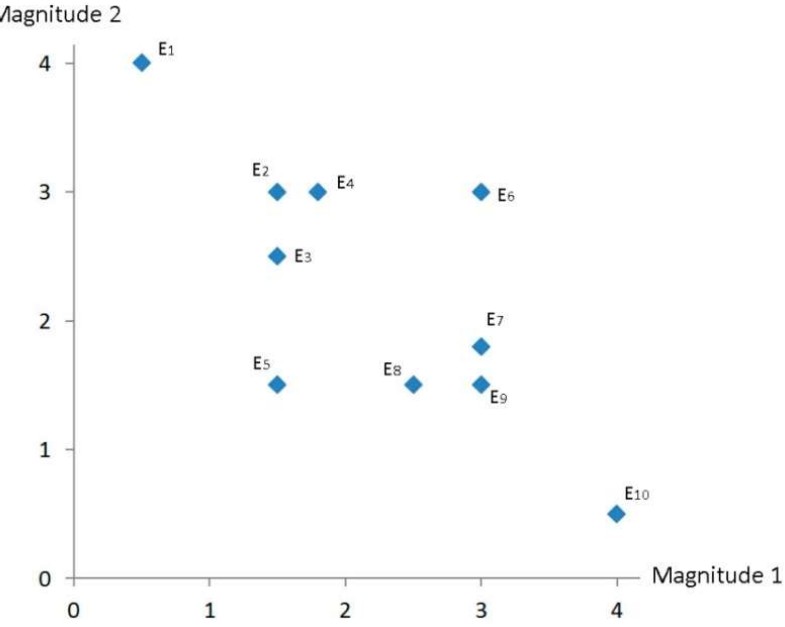

**Figure 1.** Graphical representation of the ten firms in Table 1 ($E_1$ to $E_{10}$).

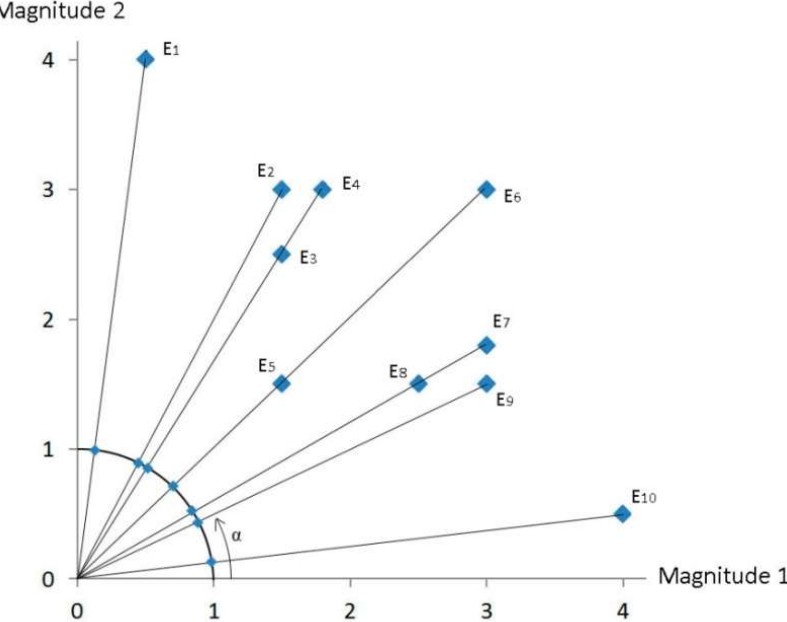

**Figure 2.** Angles ($\alpha$) of the ten firms ($E_1$ to $E_{10}$).

The column in Table 1 headed "Mg2/Mg1" contains the values of the ratio of Magnitude 2 over Magnitude 1. The same ratio values in this column for firms $E_3$ and $E_4$, firms $E_5$ and $E_6$ and firms $E_7$ and $E_8$ are justified by firms with proportional magnitudes having the same ratio.

Geometrically, the ratio of Magnitude 2 over Magnitude 1 can be interpreted as the tangent of the angle $\alpha$ formed between the abscissa axis and the ray that starts at the origin of coordinates and joins all the points with the same ratio value, as can be observed in Figure 3.

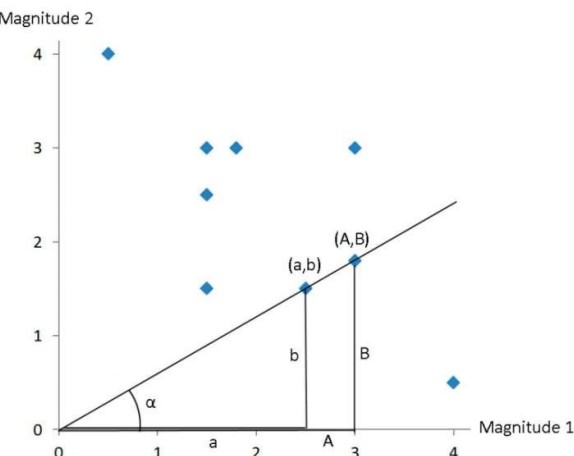

**Figure 3.** Graphical relationship between the tangent of an angle and the different points on the ray that starts from the origin of coordinates (tan($\alpha$) can be written as $\frac{b}{a}$ or $\frac{B}{A}$ ).

Consequently, given that two points (*a*,*b*) and (*A*,*B*) that share the same ray starting from the origin of coordinates always fulfill the expression *b*/*a* = *B*/*A*, in the case that *a* = 1, *b* = *B*/*A*. This fact shows that the ratio of Magnitude 2 over Magnitude 1 can be interpreted geometrically as the height of the cut-off point of the ray with the line of equation *x* = 1, as shown in Figure 4.

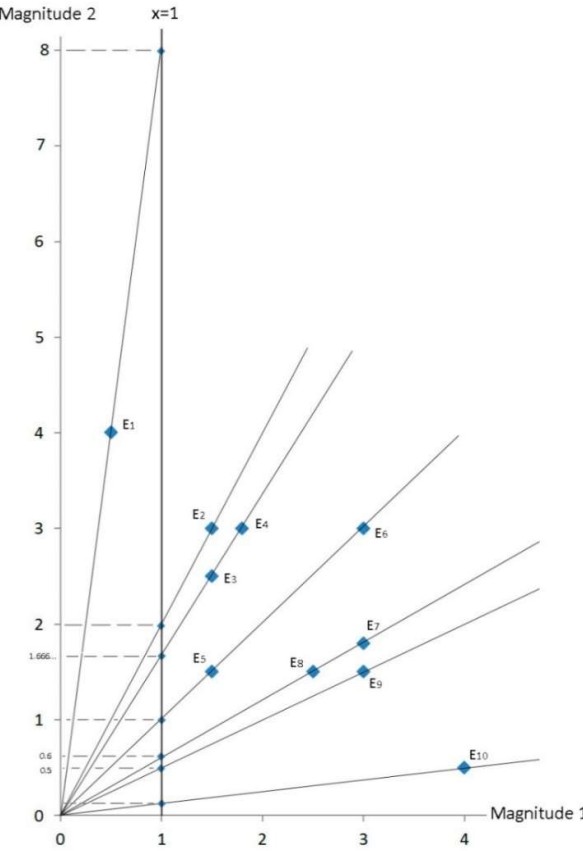

**Figure 4.** Graphical calculation of the ratio of the ten firms ($E_1$ to $E_{10}$) using the projection on the line *x* = 1.

Figure 4 is a clear example of the fact that the values of the ratios of a set of magnitudes with symmetry with respect to the bisector of the first quadrant are not symmetrical within the interval (0, +∞) but are distributed in such a way that calls into question the

validity of the results obtained in studies that have employed standard financial ratios with statistical methods.

First, the tendency to obtain positive skewness in the calculation of standard ratios can be understood immediately. The fact that the points located below the bisector of the first quadrant have a ratio value that falls within the interval (0,1), and that the points located above the bisector have a ratio value that falls within the interval (1,+∞), means that, above 1, the distribution of the ratio value necessarily has a longer tail.

Second, it is shown that the ratios do not preserve distances. Figure 4 shows that the distance between the ratios of the two magnitudes of firms $E_1$ and $E_2$ is much greater than the distance between the ratios of the magnitudes of firms $E_2$ and $E_{10}$, which is not consistent with the configuration of the data where the distance between the points on the graph representing the magnitudes between firms $E_1$ and $E_2$ is much shorter than the distance between the points representing the magnitudes between firms $E_2$ and $E_{10}$. This distortion of the distance produced by the ratio projection, proven with this simple example, shows that the results of any statistical analysis using distances with ratios, such as clustering, cannot be considered valid.

Third, the appearance of ratio values that are apparently very far from the rest, as can be seen with firm $E_1$, is a further indication of potential distortion: the ratio of firm $E_1$ can be identified as an outlier, while that of firm $E_{10}$ is not, in spite of the fact that its relative position with respect to the other firms is symmetrically the same as that of firm $E_1$. In brief, there is no reason to consider firm $E_1$ more outlying than firm $E_{10}$. The appearance of such spurious outliers is a serious practical problem when using standard ratios in statistical analysis. On the one hand, the elimination of firms erroneously identified as outliers can jeopardize the representativity of the data sample. It must be remembered that there is no objective reason to consider the firm $E_1$ as deviant—it only appears so because of the ratio value. On the other hand, not eliminating these firms, which are in fact outliers in the value of the standard ratio, is known to distort the result of most standard statistical techniques, including least squares regression analysis, analysis of variance (ANOVA), and even a simple calculation of industry means. This forces the researcher to use more sophisticated robust methods such as MM regression methods or permutation ANOVA tests.

Related to the previous point, there is another consideration related to the inconsistency of the results obtained when using the inverse ratio, interchanging the numerator and the denominator. From a mathematical standpoint, given two accounting magnitudes that are not zero, there is no reason beyond agreement to justify the choice as to which is the numerator and which is the denominator of the ratio, because the fact that the economic significance related to the times that the first magnitude contains the second that can be conferred on the value of a ratio is mathematically equivalent in the case of selecting the inverse ratio. To give an example, if we have the magnitudes 8 and 4, the value of the ratio 8/4 = 2 can be interpreted as the first magnitude being double that of the second, similarly to the case in which the value of the ratio 4/8 = 0.5 can be interpreted as the second magnitude being half of the first. In short, the same values should convey the same information. Unfortunately, this is not the case. Table 1 also contains the value of the ratio of Magnitude 1 over Magnitude 2, whose values can be interpreted geometrically as the abscissa of the cut-off point of the ray with the line of equation $y = 1$, as shown in Figure 5. The data of the firm identifiable as a possible outlier are, for this ratio, those of firm $E_{10}$ rather than firm $E_1$, leading us to consider that two researchers starting with the same firm data can obtain different conclusions. Even when outliers do not appear, the permutation of the numerator and the denominator modifies the distance between firms. With reference to Figure 5, the distance between the ratios of firms $E_1$ and $E_2$ is much smaller than the distance between the ratios of firms $E_2$ and $E_{10}$. Their skewness is also altered, and the firms previously located on the longest tail of the distribution are now located on the short one, and vice versa. This fact has special practical relevance because, in the accounting literature, it is quite common to encounter studies using inverse versions of the same ratio [49].

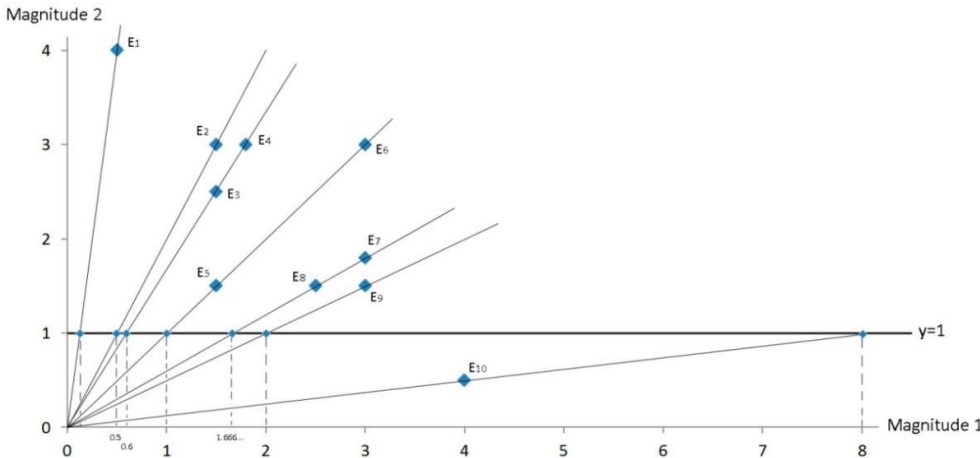

**Figure 5.** Graphical calculation of the ratios of the ten firms (E$_1$ to E$_{10}$) using the projection on the line $y = 1$. Given that two points (*a*,*b*) and (*A*,*B*) that share a line from the origin of coordinates always fulfill the expression $\frac{b}{a} = \frac{B}{A}$, if $b = 1$, then $a = \frac{A}{B}$.

## 3. Materials and Methods

### 3.1. Financial Ratios Based on the CoDa Methodology

The previous section focused on showing the serious problems that can be encountered when using standard financial ratios as data in the statistical analyses of an industry by using a simple mathematical counterexample based on simulated data. The aim of this section is to present a methodology that solves these problems. To this end, isometric log-ratio coordinates (ilr coordinates), a type of ratio used in the so-called compositional data analysis methods, or, simply speaking, Compositional Data (CoDa), are introduced. The CoDa methodology emerged from the fields of geology and chemistry, areas in which the focus of interest of the chemical analyses carried out is the relative importance of the parts of the rock or substance being analyzed, and where the size of the sample becomes irrelevant. While the CoDa methodology originated in response to the problems found when applying standard statistical methods to data of the parts of a whole, often with a constant sum [23,50], CoDa are nowadays mainly associated with the interest in relative magnitudes, to the point where [51] (p. 600) defined CoDa simply as "arrays of strictly positive numbers for which ratios between them are considered to be relevant", which fits perfectly with the analysis of financial statements using ratios.

As with standard financial ratios, the starting point when using financial ratios based on the CoDa methodology is the study objectives. For example, imagine that the aim is to review the indebtedness of firms, which could mean working with the following standard financial ratios:

$$\text{The solvency ratio: } r_1 = x_1/(x_2 + x_3), \tag{1}$$

$$\text{the debt maturity ratio: } r_2 = x_2/x_3, \tag{2}$$

and the three positive magnitudes of the following account categories: $x_1$ = total assets, $x_2$ = non-current liabilities, and $x_3$ = current liabilities. Of course, other ratios and account categories could be added. The objective is not to provide a general view of the companies' financial health but to provide a simple motivating example of the problems of standard financial ratios and how the CoDa approach tackles them.

The proposal of what the CoDa methodology calls ilr coordinates, which can be interpreted as the standard financial ratios to be used to study the problem in question, starts with a tree diagram, as shown in Figure 6, where each branch sequentially partitions the set of account categories analyzed, in progressively smaller groups until each account category is a group in itself. At each partition of the tree, the ilr coordinates, which will herein also be referred to as compositional financial ratios, position the two groups of account categories involved in the partition (or more precisely their geometric means), one

as the numerator and one as the denominator. The CoDa methodology shows that no more than two ilr coordinates will ever be needed to study the relative size of three account categories [52]. The proposal is to use the following two compositional financial ratios to study the problem at hand:

$$y_1 = \sqrt{\frac{2}{3}} \log \frac{x_1}{\sqrt{x_2 \cdot x_3}} , \qquad (3)$$

$$y_2 = \sqrt{\frac{1}{2}} \log \frac{x_2}{x_3} . \qquad (4)$$

It can be observed that the first compositional financial ratio $y_1$ contains all the account categories. A scaling factor appears, multiplying the logarithm in which the total number of account categories that intervene in the ratio $(2 + 1)$ appears in its denominator and, in its numerator, the product of the number account categories that appear in the denominator and the numerator $(2 \times 1)$. After this, the numerator and the denominator of the log ratio are simply the geometric means of the associated accounting figures if there is more than one. One should also note that in the numerator and the denominator, the assets (numerator) and liabilities (denominator) are separated, thus providing a notion of solvency, as is the case with the ratio $r_1$.

Second, the compositional ratio $y_2$ contains the two grouped account categories in the tree diagram shown in Figure 6. The scaling factor once again contains the total number of account categories $(1 + 1)$ in its denominator and the product of the number of account categories that appear in the denominator and the numerator $(1 \times 1)$ in its numerator. Note that this ratio can be interpreted as a measure of the debt maturity, comparing non-current and current liability, as is the case with $r_2$.

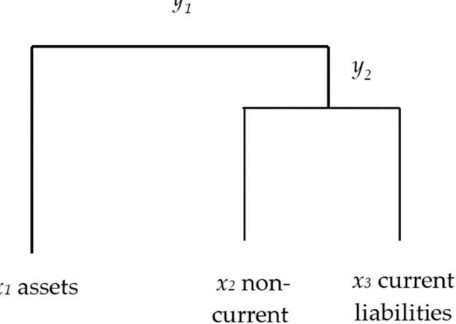

**Figure 6.** Tree diagram.

The ratio proposal presented enables us to deduce that an ilr coordinate with a positive sign indicates that the account categories in the numerator are more important than those in the denominator. From the manner in which the account categories are placed in the numerator and the denominator, we can see that a higher $y_1$ compositional financial ratio is interpreted as higher solvency, and a higher $y_2$ compositional financial ratio is interpreted as higher debt maturity.

This section ends with some important observations associated with the use of ratios in the CoDa methodology.

First, it must be pointed out that while any partition tree will be suitable, statistically speaking, it is advisable to select one that can be interpreted according to the usual concepts in financial ratio analysis. The tree in Figure 6 could be different, but the choice made here is aimed at obtaining two ratios related to the concepts of interest, solvency, and debt maturity, with a similar interpretation as the corresponding standard financial ratios.

Second, the choice regarding placement in the numerator or the denominator of the compositional financial ratio does not modify any other property of the log-ratio coordinate except for the sign. For instance,

$$\sqrt{\frac{2}{3}}\log\frac{x_1}{\sqrt{x_2 \cdot x_3}} = -\sqrt{\frac{2}{3}}\log\frac{\sqrt{x_2 \cdot x_3}}{x_1}. \tag{5}$$

This fact means that the permutation of the account categories of the numerator and the denominator ensures the following:

- The same values identifiable as outliers in both cases are obtained.
- The same skewness statistic is obtained, but with the opposite sign.
- The relationships with non-financial external variables (e.g., differences in means, correlations, coefficients of regression) are identical in size, but with opposite signs.

Third, it must be pointed out that any other possible proposal of ilr coordinates can be created from linear combinations of the previous ratios. For example, an ilr coordinate that compares the assets with the non-current liability is given by:

$$\sqrt{\frac{1}{2}}\left(\sqrt{\frac{3}{2}}y_1 - \sqrt{\frac{1}{2}}y_2\right) = \sqrt{\frac{1}{2}}\log\left(\frac{x_1}{x_2}\right). \tag{6}$$

This observation leads us to highlight the fact that, in CoDa, it is not necessary to seek new ratios that improve the result presented because, given a set of $D$ account categories, the construction of a set of $D$-1 ilr coordinates already contains all the possible information [52], and adding more coordinates will not provide more. Consequently, choosing to work with compositional financial ratios has a secondary effect, eliminating the ratio mutual redundancy encountered in studies with a large number of standard financial ratios [49], for example when using leverage and solvency ratios simultaneously.

Fourth, it can be deduced from working with logarithms that the compositional ratios have values within the entire interval $(-\infty, +\infty)$, an important fact given that the normal statistical distribution also has values within this interval and which contributes to reduce skewness.

Fifth, the scaling factors $\sqrt{2/3}$ and $\sqrt{1/2}$ are used to take into account the number of accounting magnitudes, but they do not affect the interpretation or statistical significance of any statistical analysis.

### 3.2. Example Data

The aim of the present section is to use a real data example to show the different results that can be obtained from a financial statement analysis of a particular industry using standard and compositional financial ratios. The fact of obtaining different results for the same practical case study must serve to draw researchers' attention to the methodology to be followed and to generate debate on the subject. The results of a financial statement analysis at the industry level depend on the methodology chosen, and, therefore, we are of the opinion that the new methodology presented in this article can increase the accuracy of the solvency studies of a particular industry. To this effect, this article supports the selection of the new CoDa methodology proposed, the higher validity of which is shown by means of the example that follows.

The example data used are from the Spanish wine industry. In Spain, all 17 regions (called autonomous communities) produce wine, and wineries help to prevent the depopulation of certain rural areas. In addition, wine production is an industry subject to innovation, thanks to the 70 Denominations of Origin and the 42 Protected Geographical Indications, with an internationalized wine resulting from its exports and the fact that it represents 13% of the vineyard surface worldwide. It is an important industry at both an economic and social level, as it regularly generates over 2% of the gross value added and of

jobs in Spain. This country is the world's largest exporter of wine in volume and the third in value [53].

The data were obtained from the Iberian Balance sheet Analysis System (SABI) database, developed by INFORMA D&B in collaboration with Bureau van Dijk. The search criteria were winery firms (NACE 1102 "Manufacture of wine from grape") in Spain with available data for 2016, from which a sample of the largest firms was selected covering 76.3% of the industry's total net sales ($n = 110$). We also considered the categorical variable indicating whether the firm sold at least some products using its own brand. Wineries without a brand tend to sell bulk unmature wine, while firms with a brand tend to sell aged, bottled wine, which decreases turnover and changes both the asset structure and the need for long-term debt in the capital structure.

## 4. Results

The standard and compositional financial ratios used are those presented in the previous section, $r_1$, $r_2$, $y_1$, and $y_2$, in addition to the ratios resulting from the permutation of the numerator and the denominator; specifically,

$$r_{1p} = (x_2 + x_3)/x_1, \tag{7}$$

$$r_{2p} = x_3/x_2, \tag{8}$$

$$y_{1p} = \sqrt{2/3}\ \log(\sqrt{x_2 \cdot x_3}\,/x_1), \tag{9}$$

$$y_{2p} = \sqrt{1/2}\ \log(x_3/x_2). \tag{10}$$

These inverted ratios contain the same information as the original ones ($r_1$, $r_2$, $y_1$, and $y_2$), but they are now expressed in terms of indebtedness rather than solvency, and a short maturity of the debt rather than a long maturity. In particular, $r_{1p}$ is called debt ratio and is used in standard financial statement analysis at least as often as $r_1$.

The example focuses on the comparability of graphical displays (box plots), skewness, and kurtosis, identifying any outliers present, and on the relationships between the financial statements and the external variable indicating the existence of an own brand. In this case, the external variable is categorical, and the analysis of the relationships consists of comparing the means of two groups of firms defined by the external variable using standard two-sample *t*-tests. It can also be understood as a regression analysis with a binary dummy regressor.

The box plots in Figures 7 and 8 show the standard ratios $r_1$, $r_2$, and $r_{2p}$ to have extreme asymmetry and extreme outliers. This is not the case for the compositional ratios $y_1$, $y_2$, $y_{1p}$, and $y_{2p}$. The graphs for $y_{1p}$ and $y_{2p}$ are simply the graphs for $y_1$ and $y_2$ turned upside down. Contrarily, the graphs for $r_{1p}$ and $r_{2p}$ significantly change their appearance with respect to $r_1$ and $r_2$.

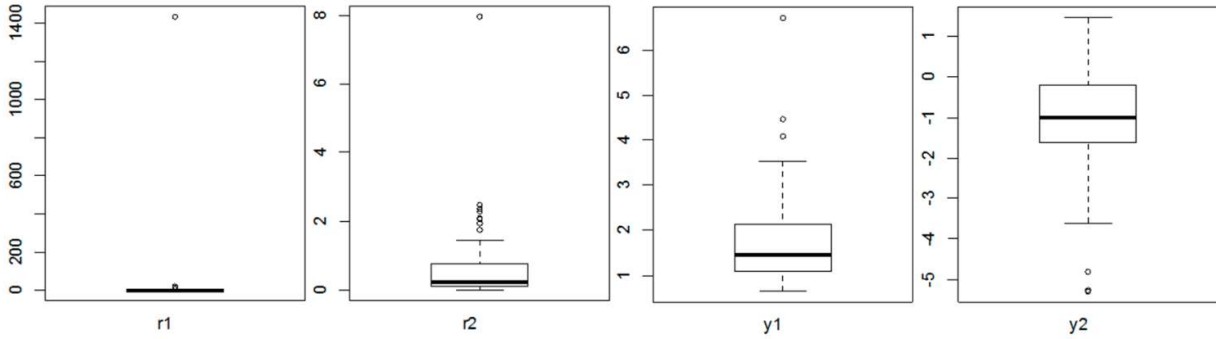

**Figure 7.** Box plots of standard ($r_1$, $r_2$) and compositional ($y_1$, $y_2$) financial ratios.

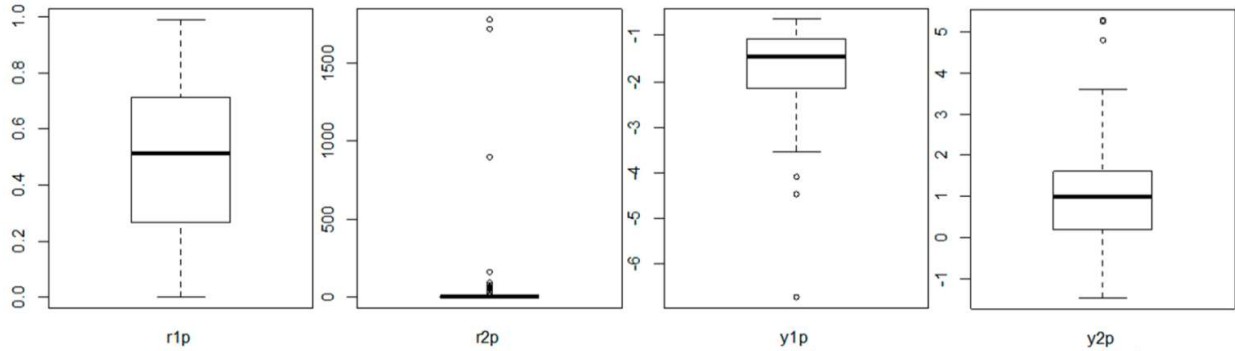

**Figure 8.** Box plots of standard ($r_{1p}$, $r_{2p}$) and compositional ($y_{1p}$, $y_{2p}$) financial ratios after permutation.

Table 2 once again shows $r_1$, $r_2$, and $r_{2p}$ to have extreme asymmetry and extreme outliers. The statistics and outliers for $y_{1p}$ and $y_{2p}$ are the same as for $y_2$ and $y_{1p}$ after reversing the skewness sign, and they reveal far lower skewness, far fewer outliers, and very rare extreme outliers.

**Table 2.** Skewness, kurtosis, and count of outliers of standard and compositional financial ratios before (top) and after (bottom) permutation.

|  | Skewness | Kurtosis | n Outliers * | n Extreme Outliers ** |
|---|---|---|---|---|
| $r_1$ | 10.48 | 109.87 | 10 | 7 |
| $r_2$ | 5.27 | 38.85 | 9 | 1 |
| $y_1$ | 1.97 | 6.53 | 3 | 1 |
| $y_2$ | −0.98 | 1.52 | 3 | 0 |
| $r_{1p}$ | −0.01 | −1.16 | 0 | 0 |
| $r_{2p}$ | 6.34 | 40.72 | 22 | 17 |
| $y_{1p}$ | −1.97 | 6.53 | 3 | 1 |
| $y_{2p}$ | 0.98 | 1.52 | 3 | 0 |

\* beyond 1.5(Q3–Q1) from the box boundary in the box plot. ** beyond 3(Q3–Q1) from the box boundary in the box plot.

In Table 3, $y_2$ and $y_{2p}$ equivalently show firms with at least one own brand to have a significantly longer debt maturity. This relationship is also marginally significant for $r_2$ but not for $r_{2p}$. $r_1$ and $r_{1p}$ are also in contradiction with respect to a relationship between solvency and having an own brand. This relationship does not emerge with $y_1$ and $y_{1p}$. The results regarding relationships with external variables thus change from standard to compositional financial ratios, and, in the former case, they also change depending on the numerator and denominator choice.

**Table 3.** Equal-variance two-sample *t*-test (positive *t*-value which means the "yes" group—having at least one brand—has a larger ratio mean than the "no" group), and R-squared (percentage of explained variance of the ratio in a dummy variable regression with having or not an own brand as predictor).

|  | *t*-Value | *p*-Value | R-Squared |
|---|---|---|---|
| $r_1$ | 0.53 | 0.59 | 0.3% |
| $r_2$ | 1.88 | 0.06 * | 3.2% |
| $y_1$ | 0.25 | 0.80 | 0.1% |
| $y_2$ | 2.14 | 0.03 ** | 4.1% |
| $r_{1p}$ | −2.23 | 0.03 ** | 4.4% |
| $r_{2p}$ | −0.78 | 0.44 | 0.6% |
| $y_{1p}$ | −0.25 | 0.80 | 0.1% |
| $y_{2p}$ | −2.14 | 0.03 ** | 4.1% |

\* Significant at 10%. ** Significant at 5%.

As expected, we find that log-ratio coordinates are much better suited for statistical analysis, having fewer outliers and less skewness and kurtosis, and that permuting the denominator and numerator does not modify the results.

On the contrary, when using standard financial ratios, permuting the numerator and denominator leads to substantial differences in the conclusions: outliers emerge or disappear, and the relationships with the external variable become significant or insignificant. The results in any case are invalid whenever extreme outliers are present.

## 5. Discussion

The main objective of this article is to show that the use of standard financial ratios in studies at the industry level is not recommended because the results produced can be invalid, even when applying the most elementary statistical techniques. The present article complements the already existing research that exposes the serious consequences of using financial ratios in multivariate statistical analyses [5,16,22,45–48], while revealing the causes of the problems and showing them with two simple examples, of which one consists of real data. In line with what was shown in this article, it must be remembered that because asymmetrical distributions can lead to the relations between the standard financial ratios being non-linear [5], using the latter can impede, for example, the application of factor analysis, regression analysis, and other linear models [46]. Another obvious example of the invalidity of results can be found in the specific case of cluster analysis. In asymmetric distributions, some of the clusters can end up being very small [54–56], which is detrimental to the grouping of firms. This problem is related to the appearance of spurious outliers, which we have already seen, and, in this case, tends to generate clusters with a single firm [22,47]. This problem is often ignored in practice even though it is very well-known in theory [5,7,19,20], causing extremely asymmetric distributions, the possible main sources of which are the aforementioned spurious outliers [21,46], a fact that could be a source of the invalidity of even the conclusions drawn from studies related to simple ratio averages [57].

The mathematical counterexample and the empirical example in the present article both show that working with financial ratios based on the CoDa methodology minimizes the problem of the appearance of outliers and solves the problem of asymmetry of information in choosing the numerator and the denominator, a well-identified problem in financial statement analysis [21].

This article shows that the construction of two CoDa ilr coordinates related to three accounting magnitudes enables the indebtedness of the wine industry in Spain to be studied. The methodology presented can be extended to the study of any $D$ accounting magnitudes with $D$-1 ilr coordinates, such that their construction can meet any other study objectives. This opens up the possibility of enormously simplifying applied research, related to the fact that it can be shown that only $D$-1 ilr coordinates include information about all the possible ratios between any two account categories of the $D$ account categories considered [27], thus preventing the redundancy problems encountered in the financial literature when a very large number of ratios are used [1,45,49]. The different ways in which the tree diagram can group the $D$ magnitudes will shape the objectives of the study. We would like to highlight that, if it were required, other positive magnitudes whose size is to be compared with accounting magnitudes in relative terms could be added. This even applies to non-monetary magnitudes such as the number of employees and other magnitudes that are typical in management, strategy evaluation, and performance ratios. The number of employees can be used to construct ratios such as the average wage, earnings per employee, or assets per employee. To this effect, the fact that there are no restrictions on the types of magnitude used must be underlined, as long as they are positive.

Another potential way to broaden this research is related to the various beneficial secondary effects produced by using financial ratios based on the CoDa methodology—for example, the improvement in the distribution of the magnitudes of the compositional ratios such that it is normal or nearly so [23,27], a characteristic rarely seen in standard ratios [8,19,20]. Another effect derives from calculating the Euclidean distance over ilr

coordinates, which is the same as working with the Aitchison distance [58] over accounting magnitudes [22,52]. This implies not having to modify the algorithms to include new distances when, for example, making classifications or data visualizations, the calculations for which can be made with the Euclidean distance over the transformations of the accounting magnitudes in the form of ilr coordinates. Revisiting current financial statement analysis studies using the CoDa methodology could be a source of much related research.

One limitation to bear in mind both when working with the CoDa methodology and with standard financial ratios derives from the mathematical assumption that ratios whose accounting figures can simultaneously be zero and positive and negative make no sense. First, a ratio with a zero numerator has no possible interpretation, so it cannot be deduced from the zero value that the numerator is 10 times smaller than the denominator, or 100 or 1000. The zero value of this ratio simply contributes no information. If the denominator is zero, we cannot even compute the ratio. The case of simultaneous positive and negative magnitudes is even more serious. It is unjustifiable that two firms with close values offer interpretations that cause researchers to believe that these very similar firms are completely different. More specifically, if we have two almost identical firms with the same numerator, but with denominators that are practically 0, one slightly positive and the other slightly negative, their values for the ratio can be as different as one wishes. Hence, comparing the values of this ratio is meaningless. What comparative information about financial health would be extracted from a value of one thousand for one firm, and minus two million for the second firm? Is the first firm so much better off in terms of financial health than the second? In this line, Ref. [59] already pointed out that computing a ratio is a meaningful operation only for variables in a ratio scale, which need to have a meaningful absolute zero and thus no negative values. Unfortunately, magnitudes such as profit, cash flow, net worth, and working capital, which can be negative and dispense with this recommendation, have become usual in financial ratios.

The assumption of non-negative accounting magnitudes is a non-negotiable assumption in CoDa and should be recognized as a non-negotiable assumption in the analysis of financial ratios as a whole. Avoiding negative accounting figures must always be recommended, converting them into positive figures, which can always be achieved without losing any information. One example is replacing the ratio of profits over assets with two strictly positive ratios, one of income over assets and the other of expenses over assets. The difference between the two ratios contains the same information as the replaced problematic ratio. In the same vein, ratios computed from current assets and current liabilities should replace ratios computed from the working capital or ratios computed from cash inflows, and cash outflows should be separated in the analysis of the cash flow statement [44]. This recommendation has already been made in the area of accounting [7] but has unfortunately generally been overlooked.

Regarding the zero-value issue, the CoDa methodology provides a diverse toolbox with imputation methods for zero values under the most common assumptions [60–62], which provides CoDa with a head advantage compared to standard financial ratio analysis in the presence of zeros.

Lastly, it must be mentioned that, in line with other CoDa methodology proposals, apart from the ilr coordinates, other types of compositional financial ratios can be used to replace standard financial ratios. These include pairwise log ratios [16] and centered log ratios [45,47,48]. The advantage of the ilr coordinates used in this article is their general applicability to any statistical method [22,26,44,46,51,52]. For example, pairwise log ratios are not applicable to statistical methods that use distances, such as cluster analysis, and centered log ratios are not applicable to statistical methods that require the inversion of covariance matrices, such as multivariate analysis of variance (MANOVA).

## 6. Conclusions

This article presents a detailed account of the fact that, while being a valid methodology for studying the financial reality of single organizations, the methodology of standard

financial ratios presents serious problems when the ratios are used as variables in statistical analyses and can produce results contaminated by spurious outliers, non-linear relationships, inconsistencies depending on the numerator and denominator choice, or data bounded to be positive, which can never follow a normal statistical distribution.

For the first time, this article shows that the cause of the abovementioned problems and other related problems is the asymmetry produced in calculating standard financial ratios. The fact of dividing two numerical values causes a distortion of the symmetry per se. A simple mathematical counterexample with simulated data was sufficient to evidence this.

This article presents a working methodology that enables the problems associated with the use of standard financial ratios to be eliminated or minimized. The methodology is based on a type of ratio used in the so-called CoDa approach. The soundness of this methodology was shown in many references, both theoretical and applied to different research fields, offering the hope that its nascent applicability to accounting can solve the current methodological problems.

The article illustrates the advantages of the CoDa approach with a toy example of only two ratios in a small sample of wine producers. The purpose of the example is only comparative, and it by no means purports to offer a complete representation of the financial structure of wineries. The example shows that the relationships with the branding variable change substantially with the CoDa approach and that the results of standard financial ratios are affected by the presence of extreme outliers and depend on arbitrary ratio permutations.

We aim at opening a dialogue between the fields of accounting and statistics, around the need to find a new working methodology to diagnose the financial health of a business activity. Given that more reliable results provide a better basis for taking economic, managerial, and financial decisions, the search for a reliable methodology must be in the current interest for the accounting scientific community. This is the main policy implication of this article.

To this effect, the present article takes the wine industry as an example for comparative studies among different methodologies to diagnose financial health in different industries. We hope, therefore, that this will encourage studies focused on any good or service aimed at obtaining more accurate financial statement analyses. These studies can use the common range of accounting figures and ratios far beyond those used in our simple illustrative example. They can also use any standard statistical method once the data are transformed as ilr coordinates.

**Author Contributions:** Conceptualization, S.L.-M. and M.À.F.-N.; methodology, S.L.-M. and G.C.; formal analysis, G.C.; resources, N.A.-S.; data curation, N.A.-S.; writing—original draft preparation, S.L.-M., M.À.F.-N. and N.A.-S.; writing—review and editing, S.L.-M. and G.C.; funding acquisition, G.C. All authors have read and agreed to the published version of the manuscript.

**Funding:** This research was funded by the Spanish Ministry of Science and Innovation (AEI/10.13039/501100011033) and by ERDF A way of making Europe, grant number PID2021-123833OB-I00; the Spanish Ministry of Health, grant number CIBERCB06/02/1002; and the Government of Catalonia, grant numbers 2017SGR656, 2017SGR386 and 2017SGR155.

**Data Availability Statement:** Data are available at the Iberian Balance Sheet Analysis System (SABI) database, accessible at https://sabi.bvdinfo.com/ (accessed on 24 September 2018).

**Conflicts of Interest:** The authors declare no conflict of interest.

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
