# Peer review of "New Financial Ratios Based on the Compositional Data Methodology"

_axioms, doi:10.3390/axioms11120694_

Round 1
Reviewer 1 Report
This study is a theoretical and mathematical effort to analyze the financial health of a group of firms. Author(s) claims that existence of asymmetry problem in financial ratios, makes the existing methods not reliable. To solve this problem this research attempts to employ a new methodology based on compositional data (CoDa) to analyze the financial statements of a sector, and try to improve the analyzes by using conventional ratios . according to them, this new estimation method is a new technique and does not depend on the arbitrary choices.
.
I have the following comments for the benefit of the author(s).
- The topic is very important in the field of finance and employing and innovation of new teqniges has significant impact in provide the most solid, correct information to the policy makers and investors and overall very mucy worth investigation.
- The sections are balanced and well presented.
- The paper lacks a coherent literature review. I suggest adding a comprehensive literature review section.
- The Models have been defined well and explained properly.
- Tables and graphs are well presenting the results, though it is recommended to transfer all tables to the end with proper numbering and quote them in the text.
- The conclusion section is too short comparing to the whole paper, and I suggest to expand this sections and include more of policy implications of this new method .
- This paper needs to be professionally written and edited and its language be a smooth English. Sentences are too long and should be short.
- My recommendation: Without any doubt, this research is extremely important in the field of finance and financial economics. Thus, the topic of this paper is a relevant and current topic with important policy implications. But I believe this paper needs to improve substantially before journal proceed it for publication. Author(s) need to make proper changes .
Author Response
Dear reviewer.
Please see the attachment.
Sincerely yours.
The authors.

Reviewer 2 Report
Dear Sir,
After review of the paper titled "New financial ratios based on the compositional data methodology"., I think that this paper is suitable for publication in the Journal axioms in the current form.
This paper presents an important and significant empirical results which can be used by academics and experts.
This paper needs especially a routine revision in the grammatical sentences.
All best
Author Response

(The authors gave the same response as above.)

Reviewer 3 Report
The paper tries to explain why standard financial ratios can cause bias in results and propose a new methodology. Although, I can't see any interest in the topic since:
1) one financial ratio doesn't give me enough information about the company's financial wealth, so we should use a set of ratios;
2) if ratios are inversed or have similar numerator or denominator, some correlation problems can appear, so we should not analyze both;
3) outliers can be corrected instead of eliminated;
4) the authors give an example of 110 companies in the wine sector, without taking into account the company's dimension which is a factor that impacts financial ratios. For instance, the capital ratio of a small company is different than that of a large-size company; the same to return, liquidity, and so on.
Therefore, I do not recommend the acceptance of the paper as it is.
The authors should stress more the limitations of the actual methodology and the relevance of this new one. Using a theoretical example to demonstrate that the actual methodology is not the correct one is not the best option in my opinion, since it does not reflect reality. Then testing with an example of the wine sector, saying that this sector is relevant for several factors, including COVID, and then analyzing the year 2016 is incorrect. We are already in 2022 and 2016 is too far. Only a small number of companies is analyzed - 110 companies, which can also cause biased results. The new methodology proposed also has several limitations, so we cannot see it as an improvement. Finally, the authors should not write a conclusion per ideas, in my opinion.
Author Response

(The authors gave the same response as above.)

Round 2
Reviewer 3 Report
The authors have made an effort to improve the quality of the work. I still recommend the authors to stress more the work contributions.
Author Response
We thank the reviewer for once more spending his or her time on our article and for the positive feedback on the changes made. Indeed to make the contribution clear is a key issue. We have added or reformulated four paragraphs in the introduction, results and conclusion sections, which are highlighted in red.
Sincerely